# Machine vs. Human: Exploring Syntax and Lexicon in German Translations, with a Spotlight on Anglicisms

**Anastassia Shaitarova, Anne Göhring, Martin Volk**
Department of Computational Linguistics, University of Zurich
{*shaita, goehring, volk*}@*cl.uzh.ch*

## Abstract

Machine Translation (MT) has become an integral part of daily life for millions of people, with its output being so fluent that users often cannot distinguish it from human translation. However, these fluid texts often harbor algorithmic traces, from limited lexical choices to societal misrepresentations. This raises concerns about the possible effects of MT on natural language and human communication and calls for regular evaluations of machine-generated translations for different languages. Our paper explores the output of three widely used engines (Google, DeepL, Microsoft Azure) and one smaller commercial system. We translate the English and French source texts of seven diverse parallel corpora into German and compare MT-produced texts to human references in terms of lexical, syntactic, and morphological features. Additionally, we investigate how MT leverages lexical borrowings and analyse the distribution of anglicisms across the German translations.

## 1 Introduction

Advanced text generation tools such as ChatGPT[1] and Machine Translation (MT) are used by millions of people every day. With the scope of human exposure to machine-generated texts ever-growing, these tools possess the potential to have an impact on natural language. The scientific community is yet to establish a research paradigm suitable for the assessment of this impact. In the meantime, we investigate generated texts and compare them to human-produced texts. In the present paper, we focus on machine translation for the German language.

Translation study scholars long established that any translation has the potential to affect the target language (TL). First, Gellerstam (1986) noticed that the translation process leaves "fingerprints" in the TL translation and named the resulting "fingerprinted" language *translationese*. The common characteristics of (human) translated text became formalized as *translation universals* or even *translation laws* (Toury, 1995; Baker, 1995). These patterns include simplification, explicitation, overall normalization, and standardization. Moreover, the source text often "shines through" (Teich, 2003) in the target text. Kranich (2014) hypothesised that these patterns persevere beyond any given translation, reappearing in texts later produced by the native TL writers. In fact, Kranich conceptualized translation as a virtual place where languages come into contact and change as a result. The severity of change is defined by many factors, including the intensity and length of exposure.

Human exposure to MT output is expected to increase, and the global MT market is steadily growing[2]. Machine-translated texts are used in almost all spheres of life, from schools (Morton, 2022), to academic publishing (Anderson, 2021), to governments (Jaun, 2019; Dalzell, 2020; Percival, 2022), and even hospitals and courts (Nunes Vieira et al., 2020; Khoong and Rodriguez, 2022; Kapoor et al., 2022). New MT engines continue to enter the market and language coverage has reached over 200 languages (Siddhant et al., 2022) and tens of thousands language pairs across all MT systems[3].

Several researchers already started to investigate the sociolinguistic impact of machine translation. For instance, MT use has been shown to have a direct and long-lasting effect on the syntactic production of language learners (Resende and Way, 2021). While producing highly fluent

---

[1]openai.com/blog/chatgpt

[2]statista.com/statistics/748358/worldwide-machine-translation-market-size

[3]State of Machine Translation 2021 report

translations, the MT output can suffer from simplification and even impoverishment (Vanmassenhove et al., 2021; Vanroy, 2021). Moreover, MT models are known to overgeneralize and amplify societal biases (Prates et al., 2020; Farkas and Németh, 2022; Troles and Schmid, 2021; Vanmassenhove et al., 2021; Hovy et al., 2020). When it comes to the analysis of commercial MT systems, however, most research focuses on the English output of Google Translate[4] with rare mentions of other translation engines (Almahasees, 2018; Aiken, 2019; Matusov, 2019; Webster et al., 2020; Hovy et al., 2020; Brglez and Vintar, 2022).

In our paper, we explore the output of three widely used engines (Google, DeepL, Microsoft Azure) and one smaller commercial system. We work with translations from English and French to German, a morphologically and syntactically complex language. We use seven different corpora (Section 2) and a battery of evaluation metrics which examine the texts on lexical, syntactic, and morphological levels (Section 3). Moreover, in Section 3.3, we scrutinize the translations from a novel angle, by looking at the distribution of anglicisms in the German texts - the process of lexical borrowing being a crucial feature of language change and evolution (Miller et al., 2020).

## 2 Data

### 2.1 Selection of test corpora

We follow three criteria in the selection of our test corpora. First, we experiment with different domains. Second, we avoid back-translation and translationese, since they interfere with evaluation metrics and might skew the results (Toral et al., 2018; Zhang and Toral, 2019; Graham et al., 2020). However, it is difficult to find parallel corpora with a clearly-marked source language.

Finally, to prevent cross-contamination of train and test data, we work with test corpora that have not been used as training data by commercial MT systems. Since the MT companies do not disclose the composition of their training corpora, we follow a common-sense assumption that all large, publicly available parallel corpora with a dated online presence have been used for MT training. Following this logic, we refrained from using Europarl, ParaCrawl, and other similar multilingual datasets. Instead, we collected seven corpora that mostly comply with our prerequisites. We describe

them in detail in the following subsections and give a general overview in Table 1.

### 2.1.1 WMT21 and WMT22

Our first logical choice of data was the test sets for the Conference on Machine Translation[5] (WMT), since they are used for the evaluation of MT systems, and therefore consciously kept out of training data. The test sets from 2021 and 2022 contain professional translations "from scratch", without back-transaltions or post-editing.

The WMT21 News Test Set[6] is a collection of online news from 2020 aligned with professional human translations (Akhbardeh et al., 2021). The original texts are collected online in English from various American, English, and Australian newspapers as well as from Al Jazeera English, allafrica.com (a news aggregation site), two Indian news sources, and euronews-en.com, a television news network headquartered in France.

The novelty of WMT22 (Kocmi et al., 2022) is that the data comes in equal parts from 4 different domains: news, e-commerce, conversation, and social media. The test set contains roughly 500 sentences for each domain. The quality of the test data is controlled manually to avoid noise and inappropriate content.

### 2.1.2 Tatoeba

Tatoeba[7] is a non-profit association which maintains an online open depository of crowd-sourced original and translated sentences in multiple languages. The downloadable set of sentences is updated every week. We selected 1777 most recent English-German pairs dating between September and December 2022. We picked only those pairs where the source English sentences are indicated as original text and translated into German by users claiming a native or high level of German.

### 2.1.3 transX

We obtained a parallel corpus of human English-German translations containing non-sensitive data from a private translation company. Despite some of the texts being featured in the company's blog, the translation memory has not been made available to the public. The corpus contains texts about translation, editing, general business, technology, and other related topics.

---

[4]translate.google.com

[5]www.statmt.org/wmt22/
[6]github.com/wmt-conference
[7]tatoeba.org

| corpus | domain | src lang | sent pairs | one2one | tokens | src-tgt | remarks |
|--------|--------|----------|-----------|---------|--------|---------|---------|
| WMT 21 | news | en | $1,002$ | 814 | $27,937$ | web-prof | – |
| WMT22 | mixed | en | $2,037$ | $1,850$ | $39,164$ | web-prof | – |
| Tatoeba | mixed | en | $1,777$ | $1,685$ | $16,285$ | crowd-crowd | trust-based |
| transX | mixed/tech | en | $1,164$ | 965 | $20,359$ | unk-prof | urls, jargon |
| Jane Eyre | classic lit | en | $8,784$ | $3,964$ | $229,283$ | prof-prof | seen by MT |
| Text+Berg | alpine texts | fr | $22,662$ | $21,353$ | $465,776$ | mixed-unk | OCR errors |
| CS Bulletin | mixed | en | $59,348$ | $54,840$ | $1,164,694$ | prof-prof | back-translated? |

Table 1: Overview of the corpora. Number of tokens is indicated for the original source sentences.

### 2.1.4 Jane Eyre

The novel Jane Eyre by Charlotte Brontë is part of the Gutenberg Project dataset. It was aligned with its German translation by András Farkas[8] and made available on OPUS. Classical literature provides certainty about the original source language, yet is counteracted by a high likelihood that it has been seen by the commercial English-German MT models during training. Published in 1847, Jane Eyre features some archaic language and spelling.

### 2.1.5 CS Bulletin

The Credit Suisse Bulletin corpus (Volk et al., 2016) is a digitized diachronic collection of texts from the world's oldest banking magazine, published by Credit Suisse[9]. The corpus contains parallel texts in German, French, Italian, and English, and covers topics pertaining to economy, culture, sport, entertainment, etc. We selected the German-English PDF subcorpus ranging from 1998 to 2017[10]. There is no proof of the source language, and we can only assume that German was the source of most articles since Credit Suisse originated in the German-speaking part of Switzerland. Therefore, the CS Bulletin corpus here mostly represents back-translated texts.

### 2.1.6 Text+Berg

Text+Berg is a diachronic corpus of Alpine texts predominantly written by Swiss mountaineers and spanning from 1864 to 2009[11] (Volk et al., 2010; Göhring and Volk, 2011). We included all French-German parallel articles published since 1957. Due to incomplete metadata, we limited our selection to articles that explicitly stated the source language as French in the German translation, such as "Aus dem Französischen von" (*[Translated] from French by*), while excluding French articles that

were translated from a language other than French, such as "Traduit de l'anglais par" (*translated from English by*).

## 2.2 Preprocessing and Translation

We translated all source texts automatically into German using four commercial MT systems: Google Translate, DeepL, Microsoft Azure, and a small private commercial MT engine specializing in German (here: mtX). The translations were performed in November 2022. As a point of reference, we provide the translation quality scores produced by COMET (Rei et al., 2020) in Table 4. This metric draws information from both source and reference texts, and captures surface and semantic similarities. We provide more conventional SacreBLEU scores (which happen to show a similar pattern) in the Appendix A.

| corpus | azure | deepl | google | mtX |
|--------|-------|-------|--------|-----|
| WMT21 | 53.51 | **57.77** | 52.50 | 49.07 |
| WMT22 | 62.06 | **64.19** | 62.24 | 58.58 |
| Tatoeba | 71.07 | **74.13** | 72.89 | 69.92 |
| transX | 59.69 | **63.18** | 59.09 | 56.82 |
| JaneEyre | 21.23 | **29.57** | 24.14 | 17.73 |
| CSBull | 68.30 | **69.52** | 68.94 | 66.78 |
| Text+Berg | 28.78 | **41.32** | 34.38 | 31.30 |

Table 2: COMET-DA_2020 scores per MT system on full-sized corpora. The best values are in **bold**.

Since both the Credit Suisse and Text+Berg corpora contain OCR errors and poor sentence alignments, we performed an additional alignment step. We identified the most probable sentence pairs using LASER margin-based sentence alignment (Artetxe and Schwenk, 2019) with a rather strict margin criterion value of 1.2. We tokenized all texts using the Spacy-UDPipe Tokenizer[12].

The tasks of syntactic comparison and automatic anglicism analysis require precise word

---

[8]farkastranslations.com/bilingual_books.php

[9]credit-suisse.com/cn/en/content-hub/bulletin.htm

[10]pub.cl.uzh.ch/projects/b4c

[11]textberg.ch

[12]github.com/TakeLab/spacy-udpipe

alignment, which is complicated in sentence pairs with a one-to-many translation. For these tasks, we created a subsection of each corpus with only one-to-one sentence alignments. Since sentence segmentation and the choice of one-to-one or one-to-many sentences differ across translations, we selected only those sentence pairs from each translation of a corpus, where the source language sentences are the same as the ones in the one-to-one human translation pairs. In other words, we made an intersection of all translation pairs (human and MT) with an anchor on the human translation. The WMT datasets contain several human references. Here, we base our filtering on the translation that exhibits the smallest number of n-to-n pairs: WMT21 - reference C and WMT22 - reference A. The number of sentences in these subcorpora can be found in Table 1.

## 3 Metrics and Findings

We used several metrics to analyze the available translations in terms of their lexical, syntactic, and morphological features.

### 3.1 Lexical analysis

**Lexical diversity** We investigated our texts with respect to lexical diversity using a variety of metrics within the BiasMT[13] tool developed by Vanmassenhove et al. (2021). We used the Type-Token Ratio (TTR) metric, which provides a general overview of lexical diversity in a text. Since TTR is known to skew results in long texts, we also employed the measure of textual lexical diversity (MTLD), which assesses the length of word sequences with a specific level of TTR (McCarthy, 2005), as well as Yule's K (Yule, 1944), which is resilient to text length fluctuations while reflecting the repetitiveness of the data.

Although the results of our investigation show higher diversity values in human translations, several MT systems produced competitively diverse translations for some of the corpora. The mtX system scored the highest TTR values on WMT21, WMT22, Jane Eyre, and transX. It scored the highest MTLD on WMT21, and WMT22. Google scored the highest Yule's I and MTLD on the Jane Eyre translation (full results in Appendix B).

**Sophistication** Another way to examine the lexical diversity of a text is to measure its sophistication. This involves measuring how much text is

---

[13]github.com/dimitarsh1/BiasMT

filled with the most and the least frequent words. A lexically diverse text usually has a lower percentage of tokens that belong to the $1,000$ most frequent words. Subsequently, there would be a larger percentage of rare and unusual words in such a text. In our experiments, the sophistication results show the same pattern as the lexical diversity metrics. Human translations prove to be most lexically diverse in all the corpora except WMT22 and Jane Eyre where mtX exhibits the highest diversity (full results in Appendix C).

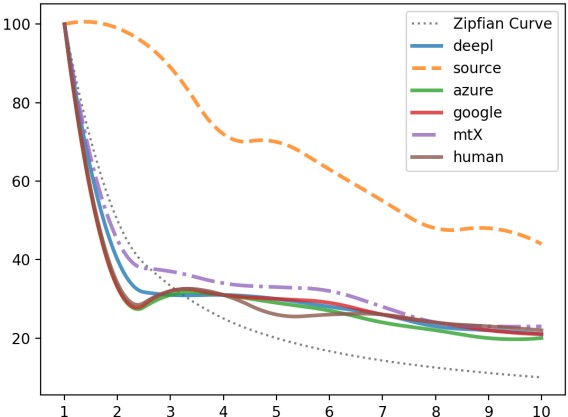

Figure 1: The Zipfian distribution of the English text and its translations in the Tatoeba corpus. The mtX output shows higher diversity of the medium frequency words than the other MT systems.

**Inflectional paradigms** Additionally, we assessed the morphological complexity and richness of each text using Shannon entropy and Simpson's diversity. Shannon entropy measures the surprisal level within each lemma's inflectional paradigm. For example, the distribution of the word forms for the German lemma *Problem* can be the following in Google's translation: {Problem:7, Probleme:3, Problemen:1, Problems:0}. If the word forms are distributed more evenly in the human translation ({Problem:4, Probleme:2, Problemen:2, Problems:3}), then the entropy for this lemma is higher than in the text translated by Google. The scores are averaged over all lemmas that appear at least as two different word forms in a corpus. Simpson's diversity reflects variability in categorical data. Higher scores indicate homogeneity, while lower scores denote diversity.

Vanmassenhove et al. (2021) observed that machine-translated English, French, and Spanish texts were less morphologically diverse than the texts used for training the same MT systems. We

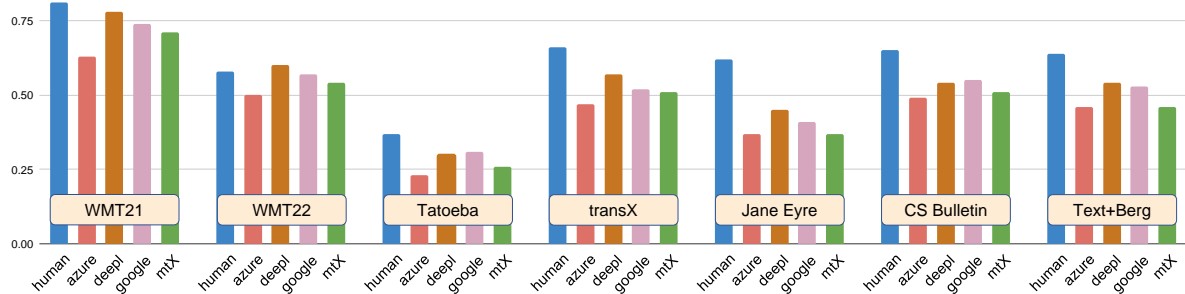

Figure 2: The measure of syntactic equivalence is calculated as the ratio of cross-alignments to the total number of word alignments. The higher score indicates more syntactically creative translation.

compare human and machine-translated texts and notice that commercial MT systems produce German texts that are comparable to human translations in terms of morphological richness. The mtX system scored higher values for the Tatoeba and the CS Bulletin corpora. DeepL produced the most diverse inflectional distributions in the translations of Jane Eyre and Text+Berg. Microsoft Azure exhibited the richest morphology in the transX corpus (see Appendix D).

**In summary,** our results show that the human translation and the MT output of the German-specialized company exhibit the highest scores for lexical diversity and sophistication. Our morphological richness results differ from the standard lexical diversity scores with more than one MT system exhibiting higher scores than the human translations.

This trend fluctuates slightly across the domains since each corpus has its own unique features. Text+Berg and CS Bulletin are large, diverse corpora with multiple writers, translators, OCR errors and specialized terminology. Tatoeba's sentences are crowd-sourced and the translators are encouraged to provide multiple translation variants. Assuming that MT tends to standardize, the lower MT diversity scores are not surprising in these corpora, although the morphological results show a different picture. The Jane Eyre and transX corpora are homogeneous in terms of domain and terminology. Here, some MT systems score higher than human texts in terms of all types of diversity.

Figure 1 illustrates lexical differences in the translations of the Tatoeba corpus using Zipf's rank-frequency distribution law. Duplicate sentences were left in for both languages. The graph demonstrates how the output of the German-specialized MT system exhibits higher diversity for mid-range frequency words, while all the translations are less diverse than the original text. Based on our results, we may infer that lexical impoverishment will not be the main issue with the machine-translated texts in the future. MT is improving rapidly for many languages, having access to more training data, and employing new decoding methods which control the diversity of the output. The quality and adequacy of translation notwithstanding, specialized systems can be tuned to produce lexically and morphologically rich texts.

### 3.2 Syntactic equivalence

We used the ASTrED tool[14] (Vanroy, 2021; Vanroy et al., 2021) to analyze the syntactic differences between texts. By dividing the number of cross-aligned words by the total number of word alignments, we obtained a measure of syntactic equivalence between the source text and its translations. The side-by-side results for all the corpora in Figure 2 clearly demonstrate that human translators exhibit greater syntactic creativity compared to any of the MT systems. These findings align with the results published by researchers for other language pairs (Tezcan et al., 2019; Webster et al., 2020; Vanroy, 2021).

Out of all our commercial MT systems, DeepL syntactically diversifies the output the most, while the other systems rather mimic the syntax of the source sentence, like in this example from the WMT21 corpus:

> **Eng**: Couple MACED at California dog park
> **Human**: Angriff mit Pfefferspray auf ein Paar in einem Hundepark in Kalifornien
> **DeepL**: Ehepaar wird in kalifornischem Hundepark angegriffen
> **Other MTs**: Paar MACED im kalifornischen Hundepark

---

[14]github.com/BramVanroy/astred

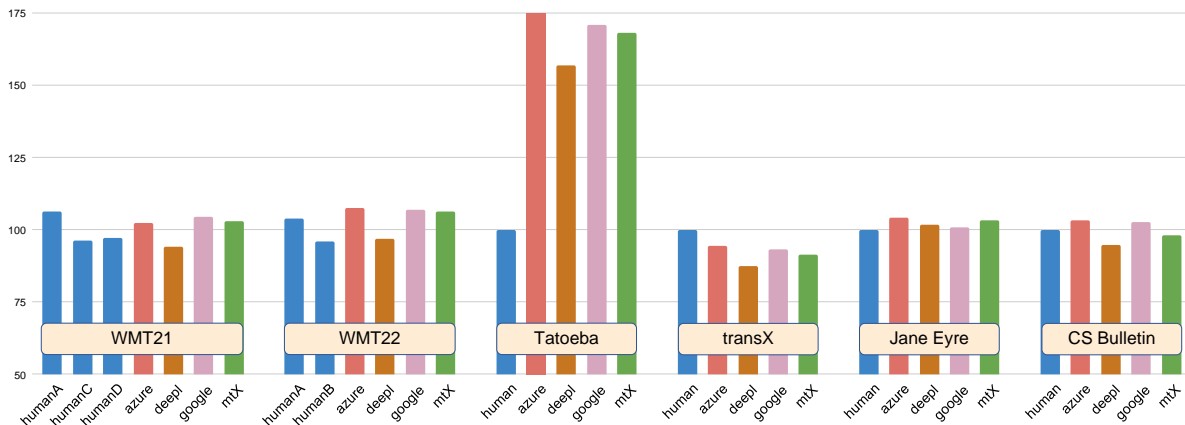

Figure 3: Distribution of anglicisms in different translations across corpora. The number of anglicisms in the human translations is taken as 100%.

Appendix E shows the translations of all 20 MT systems from the competition along with those of Google, Azure, and mtX. All of them mirror the syntax of the source sentence, whereas human translators and, to a certain extent, DeepL take liberty with the sentence structure.

### 3.3 Exploration of anglicisms

Lexical borrowings, the transfer of words from one language to another, is a productive mechanism of word formation and a catalyst of language evolution. Borrowings emerge from language contact, a universal linguistic phenomenon. They appear in all languages and can constitute a high percentage of lexical items. Identification of borrowings is important in lexicography, comparative linguistics, and some NLP downstream tasks, yet there is no reliable way to identify them automatically (Miller et al., 2020; List and Forkel, 2021).

We focus on English borrowings in German, known as anglicisms. The number of anglicisms in German is continuously growing. Reportedly, every 600th word in German could be identified as an anglicism in 1954. In 1964, it became every 200th word; in 1994, every 145th; and in 2004, every 85th (Engels, 1976; Burmasova, 2010). There is a notable societal push against this process or at least concerns about the future of the German language[15]. The investigation of this phenomenon can provide valuable insights into the role of MT in language development. We assess the extent to which MT language models participate in the an-

glicization of German. To the best of our knowledge, this is the first investigation of this kind.

There are many different ways to classify anglicisms in German: by topic, by type of surface form assimilation ("most anglicisms introduced since 1945 retain their English orthography" (Coats, 2019, p.273)), by level of assimilation (Eindeutschung), etc. Often anglicisms are classified into words indicating either new concepts (ergänzende Anglizismen, Bedürfnislehnwörter) or existing concepts (differenzierende (or verdrängende) Anglizismen[16], Luxuslehnwörter (Carstensen, 1965)). Since anglicisms continuously pour into the language but do not always stay, we work with the items that have mostly settled in German. We collected $4,832$ established anglicisms from a dedicated Wikipedia page[17], disregarding "false friends".

To avoid false positives, we filtered out certain homonyms, such as "Tag" (*day*) and "Gang" (*passageway*), and removed the word "in" which occurs in the lexicalized phrase "in sein" (*to be in*). Additionally, we excluded some corpus-specific anglicisms, for example "Credit" in the Credit Suisse Bulletin corpus, or "Miss" in the Jane Eyre corpus. The human translation of Jane Eyre contains an old, pre-1996 spelling of "Miss" as "Miß", which is not on the list of anglicisms.

We customized our search to catch different spelling variations of certain anglicisms (for example: *fairtrade, fair-trade, fair trade*). We to-

---

[15]Mind your language: German linguists oppose influx of English words; Denglisch – Deutsch oder Englisch?

[16]contify.de/glossar/richtig-schreiben/was-sind-anglizismen

[17]de.wiktionary.org: last update 12.06.2019; scraped in April, 2022

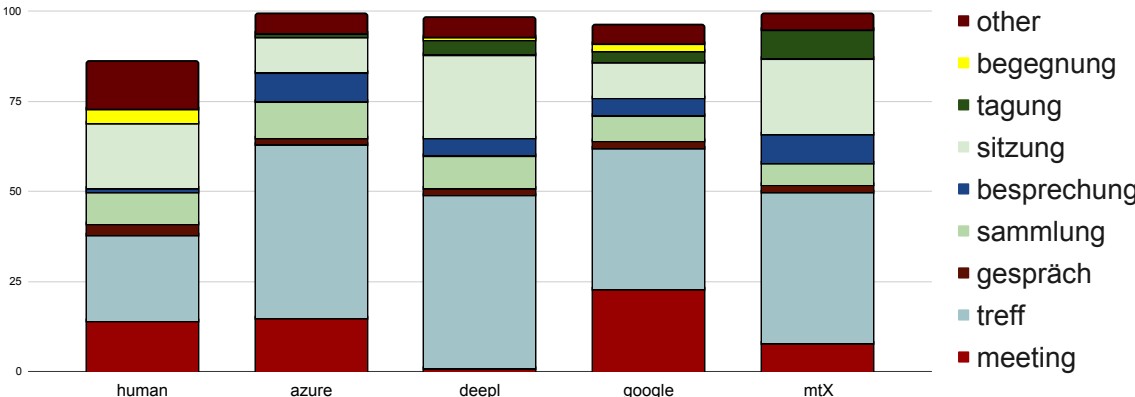

Figure 4: Distribution of lemmas for the translation variants of the anglicism "meeting" in the CS Bulletin corpus. The lemma "meeting" appears in the English text 119 times. The missing occurrences can be attributed to poor alignments.

kenized the texts with the Spacy UDpipe tool and matched anglicisms from our list to tokens, lemmas, and multiword units. Additionally, we looked for anglicisms inside German compound words. We used the Compound Split tool[18] to separate the components, and matched each component against the list of anglicisms.

We employed language detection on the produced word components to compensate for insufficient or inadequate splitting. However, language detection is not a reliable method for the identification of anglicisms. Thus, we collected the resulting alleged non-anglicisms from all the corpora into one list and manually filtered out true anglicisms. The example below shows words that were correctly and incorrectly identified as false positives of the anglicism *fan*:

> **true**: fangen, fandest, Stefan, Fannie
> **false**: Fanbasis, Autofan, Fanbild

The final list contained 342 entries, including words like *musstest* and *könntest* (falsely detected anglicism *test*); *gängig* (gig), *dadurch* (dad), *Psychologin* (gin), *hitzig* and *Hitler* (hit), etc.

Figure 3 shows the full distribution of anglicisms in all the translation versions across all corpora. The number of anglicisms in the human translations is taken as 100%. All other distributions are shown as relative to the human translation. Since the WMT corpora have several human references, the average of their scores is taken as a hundred percent mark.

While we consider the human usage of anglicisms to be the gold standard, the distributions predictably vary even among translators. Similarly, this variability occurs among the MT systems as well. Some trends are noticeable, however. For example, DeepL produces fewer anglicisms than the three other systems, while Microsoft Azure tends to anglicize its output. Figure 4 provides a distribution of translation variant lemmas for a frequent anglicism *meeting* in the CS Bulletin corpus. It shows how this anglicism barely appears in the DeepL output. Nevertheless, the overall distribution of translation variants appears to be more even in the human translation, whereas the MT systems lean towards one particular lemma (here: *treff*).

While most corpora show gentle fluctuations in the anglicism distribution across the systems, we observe a striking difference between the human and machine translations for Tatoeba. This might be due to the fact that all translations are provided by crowd-sourced volunteers, who are eager to show their love and knowledge of German. The distribution of anglicisms in this corpus has a long tail of anglicisms that were avoided by the human translators, but employed by MT: *job, meeting, online, team, internet, baby, flirt, teenager,* etc.

Conversely, the human translations of a small translation company (the transX corpus) exhibit consistently more anglicisms than the output of all other MT systems. This might have to do with the fact that professional translators follow a consistency protocol appropriate to the client's business domain (here: tech). MT systems, on the other hand, maintain a steady degree of diversification.

---
[18]pypi.org/compound-split/

## 4 Conclusion

This paper provides a corpus linguistic analysis of different translations, performed by humans and machines, in seven corpora from different domains. We looked at the texts mostly on a micro-level, measuring their lexical and syntactic properties, such as type-token ratio, morphological richness, and syntactic versatility. Additionally, we examined the distribution of translation variants for English lexical items that have entered the German language as borrowings or loan words.

Previous research emphasized that machine-produced texts suffer from standardization, simplification, and monotonicity. On one hand, our results confirm these findings in terms of syntax (section 3.2). On the other hand, we show that machine translation is becoming less of a culprit when it comes to lexical impoverishment of language. Some commercial MT systems are capable of generating German texts with levels of lexical and morphological richness similar to those produced by human translators (Section 3.1). Of course, these results reflect only one aspect of translation quality, and our automatic scores - as imperfect as they are - suggest that DeepL, not mtX, is the most reliable system for German translations (see Table 4).

Finally, we note that the standard lexical and syntactic metrics might be getting less informative for the linguistic assessment of MT as the technology continues to improve. Alternatively, automatic evaluation of lexical borrowings, such as anglicisms in German, can provide a good opportunity to assess the appropriateness of MT use. The distribution of borrowings is directly related to the quality and purpose of translation. Our results indicate that certain machine translation systems tend to produce fewer anglicisms compared to other systems (Section 3.3). In general, human translators adjust the use of anglicisms according to the domain, while the MT systems produce mostly consistent, system-specific distributions.

As machine translation improves and becomes more widespread, it will likely play a role in the (de-)anglicization of German. To mitigate this impact on German, more research is needed to accurately identify linguistic borrowings. Overall, our study sheds light on the current state of machine translation, laying the groundwork for investigating the potential impact that generated texts might have on human language.

## Acknowledgments

This research was funded by the National Centre of Competence in Research "Evolving Language", Swiss National Science Foundation (SNSF) Agreement 51NF40_180888

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

## Appendix A    SacreBLEU scores

| corpus | azure | deepl | google | mtX |
|---|---|---|---|---|
| WMT21 | 59.0 | **69.9** | 58.5 | 53.1 |
| WMT22 | 50.7 | 48.5 | **52.3** | 47.1 |
| Tatoeba | 40.6 | **42.0** | 41.8 | 39.7 |
| transX | 33.0 | **36.7** | 33.1 | 32.0 |
| JaneEyre | 18.7 | **20.1** | 19.5 | 18.6 |
| CSBull | 31.5 | **32.2** | 31.7 | 30.1 |
| Text+Berg | 23.5 | **27.2** | 24.4 | 24.6 |

Table 3: SacreBLEU scores v 2.2.1 across full-sized corpora per MT system. The best values are in **bold**.

## Appendix B    Lexical richness scores

| | system | TTR | Yule's | MTLD | | system | TTR | Yule's | MTLD |
|---|---|---|---|---|---|---|---|---|---|
| WMT21 | humanA | 24.1 | **610.79** | 134.47 | Tatoeba | human | **21.37** | **333.36** | **82.49** |
| | humanC | 23.25 | 533.61 | 125.93 | | azure | 20.39 | 274.15 | 72.49 |
| | humanD | 23.83 | 562.83 | 129.73 | | deepl | 20.27 | 276.48 | 69.29 |
| | azure | 22.84 | 456.32 | 129.14 | | google | 20.82 | 288.72 | 73.28 |
| | deepl | 22.89 | 475.89 | 127.37 | | mtX | 20.92 | 283.91 | 71.02 |
| | google | 23.06 | 488.02 | 129.74 | transX | human | 22.8 | **624.76** | **138.32** |
| | mtX | **24.13** | 528.95 | **136.34** | | azure | 22.19 | 504.05 | 132.47 |
| | | | | | | deepl | 22 | 498.44 | 130.55 |
| WMT22 | humanA | 19.17 | 369.49 | 109.5 | | google | 22.73 | 548.7 | 136.45 |
| | humanB | 19.76 | **405.13** | 111.3 | | mtX | **22.99** | 552.01 | 136.65 |
| | azure | 19.25 | 349.45 | 113.48 | CS Bulletin | human | **7.8** | **69.38** | **279.39** |
| | deepl | 19.3 | 360.71 | 110.55 | | azure | 6.55 | 46.88 | 249.45 |
| | google | 19.7 | 379.68 | 112.81 | | deepl | 6.44 | 44.84 | 228.18 |
| | mtX | **20.3** | 395.08 | **117.8** | | google | 6.88 | 54.13 | 258.3 |
| | | | | | | mtX | 6.7 | 48.75 | 249.33 |
| Jane Eyre | human | 8.08 | 59.31 | 126.88 | Text+Berg | human | **9.5** | **91.95** | **276.93** |
| | azure | 8.09 | 54.45 | 136.64 | | azure | 8.1 | 58.47 | 201.06 |
| | deepl | 8.15 | 52.06 | 127.14 | | deepl | 8.37 | 67.3 | 203.15 |
| | google | 8.38 | **63.31** | **136.87** | | google | 8.62 | 70.22 | 212.04 |
| | mtX | **8.56** | 58.01 | 129.51 | | mtX | 8.07 | 56.65 | 191.42 |

Figure 5: Lexical richness measured with Type-Token Ratio (TTR), reversed Yule's K (Yule's I), and the Measure of Textual Lexical Diversity (MTLD) across all corpora. Higher scores (in **bold**) indicate higher lexical richness.

# Appendix C  Lexical frequency profile

| group | system | B1 ↓ | B2 | B3 ↑ | group | system | B1 ↓ | B2 | B3 ↑ |
|---|---|---|---|---|---|---|---|---|---|
| WMT21 | humanA | **72.43** | 8.49 | **19.08** | Tatoeba | human | **67.72** | 6.39 | **25.9** |
| | humanC | 73.51 | 8.27 | 18.23 | | azure | 69.37 | 6.51 | 24.12 |
| | humanD | 72.77 | 8.48 | 18.75 | | deepl | 70.03 | 6.41 | 23.55 |
| | azure | 74.11 | 8.37 | 17.52 | | google | 68.73 | 6.52 | 24.75 |
| | deepl | 74.1 | 8.26 | 17.64 | | mtX | 68.81 | 6.55 | 24.64 |
| | google | 73.75 | 8.41 | 17.84 | transX | human | **78.13** | 8.88 | **12.99** |
| | mtX | 72.97 | 8.52 | 18.51 | | azure | 79.41 | 8.35 | 12.24 |
| | | | | | | deepl | 79.5 | 8.31 | 12.19 |
| WMT22 | humanA | 75.65 | 7.65 | 16.7 | | google | 78.76 | 8.58 | 12.65 |
| | humanB | 75.12 | 7.55 | 17.33 | | mtX | 78.61 | 8.52 | 12.87 |
| | azure | 75.62 | 7.63 | 16.75 | CS Bulletin | human | **83.34** | 7.53 | **9.14** |
| | deepl | 75.57 | 7.52 | 16.91 | | azure | 84.2 | 7.73 | 8.07 |
| | google | 75.1 | 7.61 | 17.29 | | deepl | 84.12 | 7.66 | 8.22 |
| | mtX | **74.6** | 7.66 | **17.74** | | google | 83.84 | 7.81 | 8.35 |
| | | | | | | mtX | 83.85 | 7.86 | 8.3 |
| Jane Eyre | human | 79.07 | 5.95 | 14.98 | Text+Berg | human | **71.55** | 6.19 | **22.25** |
| | azure | 79.7 | 5.65 | 14.65 | | azure | 73.61 | 6.13 | 20.25 |
| | deepl | 80.06 | 5.48 | 14.46 | | deepl | 73.41 | 6.04 | 20.55 |
| | google | 79.5 | 5.58 | 14.92 | | google | 72.85 | 6.15 | 21 |
| | mtX | **79.02** | 5.71 | **15.27** | | mtX | 73.97 | 5.97 | 20.06 |

Figure 6: Lexical frequency profile with B1 indicating top 1000 most frequent words, B2 1000-2000 top frequent words and B3 all the other words.

# Appendix D  Morphological richness scores

| group | system | H ↑ | D ↓ | group | system | H ↑ | D ↓ |
|---|---|---|---|---|---|---|---|
| WMT21 | humanA | **85.56** | **47.05** | Tatoeba | human | 86.74 | 47.52 |
| | humanC | 83.16 | 48.41 | | azure | 86.17 | 47.9 |
| | humanD | 84.38 | 47.82 | | deepl | 87.77 | 47.59 |
| | azure | 82.75 | 48.32 | | google | 87 | 47.55 |
| | deepl | 83.48 | 48.1 | | mtX | **88.29** | **46.93** |
| | google | 83.29 | 48.11 | transX | human | 80.21 | 49.82 |
| | mtX | 82.85 | 48.15 | | azure | **80.57** | **49.45** |
| | | | | | deepl | 79.72 | 49.86 |
| WMT22 | humanA | **82.79** | **48.98** | | google | 80.14 | 49.89 |
| | humanB | 82.63 | 49.02 | | mtX | 79.22 | 49.93 |
| | azure | 82.3 | 49.44 | CS Bulletin | human | 82.72 | 50.38 |
| | deepl | 81.33 | 50 | | azure | 86.12 | 49.04 |
| | google | 81.48 | 49.7 | | deepl | 85.01 | 49.45 |
| | mtX | 82.34 | 49.26 | | google | 85.47 | 49.33 |
| | | | | | mtX | **86.25** | **48.98** |
| Jane Eyre | human | 85.87 | 48.5 | Text+Berg | human | 84.36 | 49.41 |
| | azure | 86.82 | 48.06 | | azure | **85.79** | 49 |
| | deepl | **87.69** | **47.65** | | deepl | 85.69 | **48.81** |
| | google | 86.46 | 48.25 | | google | 84.65 | 49.47 |
| | mtX | 85.9 | 48.32 | | mtX | 84.65 | 49.46 |

Figure 7: Morphological richness measured with Shannon entropy (H) and Simpson's diversity (D). Higher H and lower D indicate morphologically richer text (marked in **bold**).

## Appendix E    Syntactic Equivalence

| human or MT | translation |
| --- | --- |
| eng | Couple MACED at California dog park |
| human1 | Paar in Hundepark in Kalifornien mit Pfefferspray besprüht |
| human2 | Paar bekommt beim Mittagessen in einem Hundepark Pfefferspray ins Gesicht gesprüht |
| human3 | Angriff mit Pfefferspray auf ein Paar in einem Hundepark in Kalifornien |
| Online-W | Paar MACED in Kalifornien Hundepark |
| Online-G | Paar MACED im California Dog Park |
| nuclear_trans | Paar MACED bei California Dog Park |
| ICL | Paar MACED bei California Hund Park |
| VolcTrans-GLAT | Paar MACED in Kalifornien Hundepark |
| P3AI | Paar Maced im kalifornischen Hundepark |
| eTranslation | Paar MACED im kalifornischen Hundepark |
| WeChat-AI | Paar MACED im kalifornischen Hundepark |
| Manifold | Paar MACED im kalifornischen Hundepark |
| VNVIDIA-NeMo | Paar MACED im kalifornischen Hundepark |
| BUPT_rush | Paar MACED im kalifornischen Hundepark |
| Online-A | Paar MACED im kalifornischen Hundepark |
| Online-Y | Paar MACED im kalifornischen Hundepark |
| Online-B | Paar MACED im kalifornischen Hundepark |
| HuaweiTSC | Paar MACED im kalifornischen Hundepark |
| UEdin | Paar MACED im kalifornischen Hundepark |
| UF | Paar MACED im kalifornischen Hundepark |
| happypoet | Paar MACED im kalifornischen Hundepark |
| Facebook-AI | Paar MACED im kalifornischen Hundepark |
| VolcTrans-AT | Paar zerfleischt im kalifornischen Hundepark |
| Google | Paar MACED im kalifornischen Hundepark |
| DeepL | Ehepaar wird in kalifornischem Hundepark angegriffen |
| Azure | Paar MACED im kalifornischen Hundepark |
| mtX | Paar MACED im kalifornischen Hundepark |

Table 4: The first clause of the first sentence in the WMT21 test set in the original English and its German translations, performed by 3 human translators and 20 participating MT systems. The bottom section of the table contains the same clause translated with the commercial MT systems for this paper.