# OpenReview forum: "Machine vs. Human: Exploring Syntax and Lexicon in German Translations, with a Spotlight on Anglicisms"
_NoDaLiDa/2023/Conference — NoDaLiDa 2023_

### Official Review · Reviewer_V5WZ · 2023-02-21
**An interesting paper comparing humand and machine translation**

**Rating:** 8
**Confidence:** 4

**Review:**

The present paper deals with a linguistic comparison of human and machine translation (MT). It is surprising that the title promise to report on the impact machine translation has on German in general: in fact, the authors compare a human (or several human) translation(s) with the outputs of several MT systems. The comparison is done on the basis of some linguistic properties of texts. These properties are inspired by works on translationese, i.e. specific features of translated languages that distinguish it from non-translation language production. However, no comparison with non-translated language (neither source nor target) is performed. Instead, several translations are compared between each other and the analysis shows that machine translated outputs have comparable properties with human translations. In general, it is an interesting piece of research with many useful insights both for NLP community and translation studies. Specifically, the analysis of anglicisms of great value. However, there are some flaws and unclear parts in the paper that should be improved for the final version.
First of all, as already mentioned above, translationese analysis requires also comparison with non-translated texts. Most of the linguistic properties the authors used can be analysed cross-linguistically. For instance, is lexical diversity of the translations influenced by the source, i.e. what is observed can be classified as 'shining through' (see Teich, 2003) or is it rather a standardisation/normalisation towards what is typical for the target language? For instance in 535ff, the authors claim that MT systems mimic the syntactic structure of the source - but how do they know this without analysing the sources?
I believe that the analyses can be left as they are as they already provide interesting insights, but the whole 'story' should be interpreted into the one on 'how MT resemble/not resemble human translations' instead of how MT impacts the language. The latter is rather an interpretation as no analysis of the target language (original German production) has been performed. If the authors add the comparison of translations with non-translated German (and it is possible with the given feature set provided there is comparable data in German), such statements can be kept.

In the following I have more specific comments on the paper content:
*Data and features:
- the human translations used int h analysis vary across subcorpora. However, variation in the translator profiles such professional, novice as well as crowd-sourced (were the WMT translations not crowd-sourced) or even back translations (!) will have an impact on the distribution of linguistic properties under analysis (see works by Corpas Pastor et al. 2008, De Sutter et al. 2017, Popovic, 2020, Kunilovskaya and Lapshinova 2020 and others who used partly similar features to analyse differences between human translator groups and also between human and machine translations).
- The analysis of sophistication can be strongly influenced by domain, register, genre diversity in the dataset (it is one of the features used for register diversification).
- It is known that translators (and especially interpreters) would use more cognates to facilitate translation process (see ). Some of the anglicism may belong to the category of cognates and be used instead of other equivalents in German. Knowing this, one would expect to have more anglicism in crowd-sourced and novice translations by humans. Could this explain some of the results the authors obtained?
- Is it correct to address Google Translate and DeepL etc. as commercial systems or better use open source instead?

*Related work:
- The work by Teich (2003) should be cited when mentioning 'shines through' (077).
- I am not sure if the lines in 081ff can be attributed to S. Kranich or rather to Juliane House (e.g. House 2015).
- There are other studies on comparing human and machine translations from a linguistic point of view (partly using similar features) which were not cited here - Lapshinova-Koltunski 2015 and 2017. However, the studies are older and therefore do not take into account neural MT.

*Results:
It would be interesting to correlate somehow the numbers obtained for the MT systems in terms of ling. features and BLEU /COMET scores.

*Formatting remarks:
- Table 1 - it is better to align numbers in a different way

References
Corpas Pastor, G., Mitkov, R., Afzal, N., and Garcia-Moya, L. (2008). Translation universals: do they exist? a corpus-based and nlp approach to convergence. In Proceedings of the LREC-2008 Workshop on Building and Using Comparable Corpora, pages 1–7.

De Sutter, G. Bert Cappelle, Orphée De Clercq, Rudy Loock, and Koen Plevoets (2017). Towards a corpus-based, statistical approach to translation quality: Measuring and visualizing linguistic deviance in student translations. Linguistica Antverpiensia, New Series–Themes in Translation Studies, 16

Heilmann, A. and C. Llorca-Bofí (2021). Analyzing the effects of lexical cognates on translation properties: A multivariate product and process based approach. In Explorations in Empirical Translation Process Research, pages 203–229. Springer.

House, J. (2015). Translation quality assessment: Past and present. Routledge.

Kunilovskaya, M. and Lapshinova-Koltunski, E (2020). Lexicogrammatic translationese across two targets and competence levels. In Proceedings of the Twelfth Language Resources and Evaluation Conference, pages 4102–4112, Marseille, France. European Language Resources Association.

Lapshinova-Koltunski, E. (2017). Cohesion and Translation Variation: Corpus-based Analysis of Translation Varieties. In Menzel, K., E.
Lapshinova-Koltunski and K. Kunz, (eds.). New Perspectives on Cohesion and Coherence: Implications for Translations. Translation and Multilingual Natural Language Processing, volume 6. Berlin: Language Science Press, p. 103-128.

Lapshinova-Koltunski, E. (2015). Variation in Translation: Evidence from corpora. In C. Fantinuoli and F. Zanettin. New directions in corpus-based translation studies.Translation and Multilingual Natural Language Processing (TMNLP), pp. 79-99. Language Science Press.

Maja Popovic (2020). On the differences between human translations. In Proceedings of the 22nd Annual Conference of the European Association for Machine Translation, pages 365–374, Lisboa, Portugal. European Association for Machine Translation.

Teich, Elke (2003). Cross-linguistic variation in system and text: A methodology for the investigation of translations and comparable texts
Berlin-New York: Mouton de Gruyter


**Paper Type:**

Long paper

---

### Official Review · Reviewer_ukPk · 2023-03-08
**Valuable data but not for the stated aim**

**Rating:** 5
**Confidence:** 4

**Review:**

This paper aims to say something significant about the impact that MT may have on human language, specifically German. It uses a corpus-based method and compares a set of metrics on seven different corpora for four MT systems and human references. The metrics concern four different aspects of text quality: lexical variation, morphological variation, syntactic variation and the distribution of anglicisms in the translations. The numbers reported are mostly averages across the corpora. Together the reported figures tell us that the corpora differ a lot as to the translation difficulty for the MT systems, that, on the whole, MT is closing in on human translation, and the order of the systems on this scale of proximity to human translation. Although these tendencies are not novel, I would have preferred the paper to focus on them and perhaps analysed them more closely.

The paper can draw no definite conclusion as regards impact. First no attempt is made to compare MT with other influencing factors, such as human translation, exposure to English through music, media, travel, work encounters etc. Second, no attempt is made to assess  the translations qualitatively. From the numbers reported on COMET and BLEU one can infer that at least two of the corpora (Jane Eyre and Text+Berg) are bound to have errors in them. A guess is that texts with errors will have less impact on readers than texts of high quality.

The study is careful in selecting test data that is likely not to have been used for training the systems. It is not clear, and not, explained what the effect would be of using open data for translation. At least it could be interesting to compare results for the two conditions.

There is quite a lot of interesting data in the paper, but it fails to reach its stated aim and for this reason I rate quality and significance low. The study on anglicisms is worth developing and the clarity of the paper is good.

**Paper Type:**

Long paper

---

### Official Review · Reviewer_6Vb6 · 2023-03-12
**The paper provides a detailed comparative analysis of German texts produced by human translators and commercial engines.**

**Rating:** 7
**Confidence:** 4

**Review:**

General feedback: This is a valuable and straightforward paper that presents insightful research findings. The authors correctly point out that while MT is widely used, it can have drawbacks that impact natural language and human communication, so it's important to evaluate MT-generated translations regularly. While the results are limited to German, the methodology can be applied to other languages. The paper would benefit from comparing human and machine translations directly and from including more macro-level analysis. Overall, the study shows that DeepL is the best MT system for German translations.

Open questions:

Is anglicism a problem of machine translation, underlying data, or something else?
Positives:

1. The paper uses appropriate and interesting datasets for the research, which adds to its merit.
2. The figures are well-presented and informative.
3. The conclusion offers valuable knowledge for MT system users - for German use DeepL.

Negatives:
1. The conclusion is overly long and could be condensed.
2. The appendix could be shortened; some pages are mostly empty.
3. I am not sold on the issue of anglicisms, which may not be as important as they suggest.

**Paper Type:**

Long paper

---

### Decision · Program_Chairs · 2023-03-17

Accept